# Experimental Study on the Preparation of Cementing Materials by Direct Reduction Coupling of a Hematite-Carbon Base

**Ruimeng Shi [1,\*], Junxue Zhao [1], Xiaoming Li [1], Chong Zou [1], Yaru Cui [1] and Guibao Qiu [2,\*]**

[1]   School of Metallurgical Engineering, Xi'an University of Architecture and Technology, Xi'an 710055, China; Zhaojunxue1962@126.com (J.Z.); xmli@xauat.edu.cn (X.L.); zouchong985@163.com (C.Z.); yaroo@126.com (Y.C.)

[2]   College of Materials Science and Engineering, Chongqing University, Chongqing 400044, China

\*   Correspondence: shiruimeng@163.com (R.S.); qiuguibao@cqu.edu.cn (G.Q.); Tel.: +86-180-0918-5550 (R.S.); +86-139-8391-2792 (G.Q.)

**Abstract:** The reduction of iron in hematite and process coupling of cementing material generated from gangue components are explored in this paper, and a technical proposal for preparing directly reduced iron and cementing materials considering the processes of energy and material flows is proposed. An experimental study preparing cementing materials, such as tricalcium silicate and dicalcium silicate, by roasting the components, was performed. In this study, hematite was used as the raw material and powdered carbon was added, as the reducing agent, with CaO; at the same time, the gangue components of iron ore were used as the principal raw materials for the process of directly reduced iron preparation by direct reduction of iron ore. The FactSage software package was used to perform thermodynamic calculations of the reduction of iron and its influence on the formation of tricalcium silicate and dicalcium silicate. The feasibility of the direct reduction of iron to elemental iron and preparation of cementing materials by roasting of gangue components under the studied thermodynamic conditions was discussed. Different temperature control strategies were used to verify the reaction coupling test. The results showed that zero-valent iron could be produced by roasting and reducing hematite under certain experimental conditions, and cementing materials, such as tricalcium silicate and dicalcium silicate, could be produced simultaneously by reacting the gangue components with CaO. $Fe_2O_3$ exerted an adverse effect on the formation of tricalcium silicate, and sufficient reduction of the iron was a precondition for the formation and stability of tricalcium silicate.

**Keywords:** hematite; directly reduced iron; cementing materials; tricalcium silicate $C_3S$ (Alite); dicalcium silicate $C_2S$ (belite); process coupling; non-slag smelting

## 1. Introduction

Direct reduction iron smelting, a new ironmaking technology, has the following characteristics: no metallurgical coke, less environmental pollution, and simple protocols. It has a higher value, especially for processing low-grade, complex, symbiotic iron ore and metallurgical dust and sludge [1–8]. However, as with traditional blast furnace ironmaking technologies, the gangue components produced by the fine-grinding process of iron mining have not been applied to reasonable and high value-added utilizations as energy carriers, resulting in energy waste. Fully utilizing the component and energy-carrying properties of gangue components after fine grinding during iron mining and realizing the high value-added resource utilization of gangue components under the reduction conditions of iron are important means to achieving low-energy, efficient, green metallurgical production. Because of

the component and energy-carrying characteristics of gangue, it is possible to prepare the gangue components into Portland cement clinker at the same time as reducing the iron. Firstly, the gangue components are the same as the main raw materials of cement clinker. Secondly, the production of cement clinker involves numerous processes and technologies that are similar to direct iron reduction and extraction technologies: a rotary kiln is used as the production equipment; the main production processes are grinding, material mixing, and material roasting and grinding; the operating temperature of the equipment is around 1300 °C; and lime needs to be added [9–13]. It is a new approach to energy conservation, emission reduction, non-slag smelting, and high-efficiency resource utilization to realize the preparation of directly reduced iron and cementing materials through energy and material flow processes and the coupling of the generation processes of the reduced iron and cementing materials. It is also a novel, efficient method for comprehensive utilization of hematite, providing technical support for the low-cost production of directly reduced iron; efficient, comprehensive utilization of resources; energy-savings; and cost reductions.

Coupling the reduction of iron in hematite and formation of cementing materials from gangue components is explored in this study. Using hematite as the principal raw material, the FactSage software package was used to perform a thermodynamic calculation analysis of the reduction of iron and its influence on the formation of tricalcium silicate and dicalcium silicate. Experimental verification was performed.

## 2. Raw Material Analysis and Composition

The hematite used in this study was a concentrated powder supplied by a company in eastern China. For X-ray diffraction analysis, the hematite powder was ground into particles smaller than 200 mesh (74 μm). The X-ray diffractometer (X'Pert PRO, PANalytical, The Netherlands) was used, and the scan condition was 8°/min; the analysis results are shown in Figure 1. At the same time, the chemical composition of the hematite powder was analyzed by X-ray fluorescence spectroscopy (ZSX, PrimusII, Rigaku, Japan), the results of which are given in Table 1.

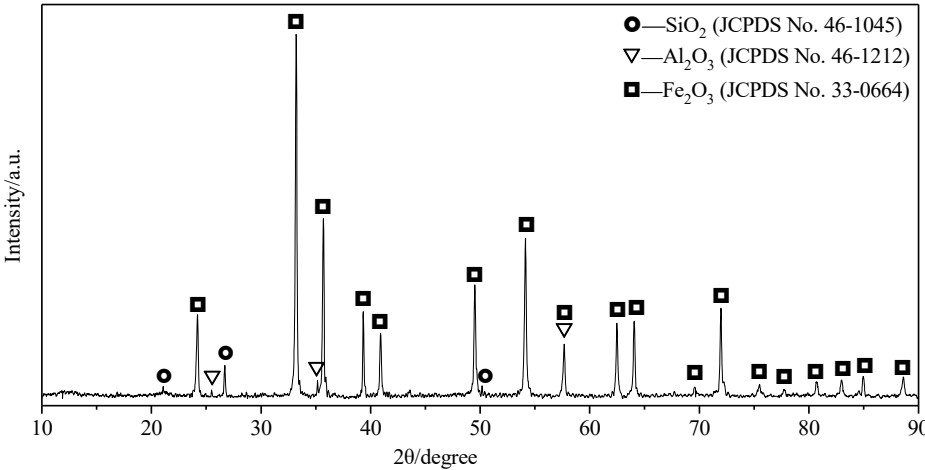

**Figure 1.** XRD pattern of hematite.

**Table 1.** Chemical composition of the hematite powder (wt.%).

| Component | CaO | SiO$_2$ | MgO | Al$_2$O$_3$ | TiO$_2$ | MnO | P$_2$O$_5$ | FeO | Fe$_2$O$_3$ | TFe |
|---|---|---|---|---|---|---|---|---|---|---|
| Content | 0.09 | 3.37 | 0.02 | 1.27 | 0.06 | 0.05 | 0.131 | 0.34 | 93.49 | 65.72 |

From the analysis of the X-ray diffraction (XRD) pattern, it can be seen that the main phase present in the hematite powder was $Fe_2O_3$. There were low X-ray diffraction peaks corresponding to $SiO_2$ and $Al_2O_3$ phases; X-ray diffraction peaks arising from minerals containing elements such as Ca, Mg, Ti, Mn, S, and P were not evident.

From Table 1 (values of the analysis of the fluorescence spectrum), it can be seen that the main phase present in the hematite powder was $Fe_2O_3$, and that the total iron content was 65.72%. There were small amounts of $SiO_2$ and $Al_2O_3$ impurities as well; the $SiO_2$ content was 3.37% and the $Al_2O_3$ content was 1.27%. There were a few other elements, such as Ca, Mg, Ti, Mn, S, and P. The process coupling of iron reduction and cementing materials generated by reaction of $SiO_2$ and CaO was calculated via thermodynamic calculation analysis, in accordance with the composition of the hematite powder. Thermodynamic calculation analysis was not conducted for the other elements, due to their low concentrations.

Based on the above analysis, the raw materials were apportioned before the experiment. The apportioning included the following two considerations: (1) the reducing agent added should completely reduce the iron; (2) the amount of CaO added should ensure sufficient combination of the $SiO_2$ and $Al_2O_3$ for the generation of tricalcium silicate and tricalcium aluminate. The raw material apportioning calculated for the reduction roasting of hematite is shown in Table 2. All raw materials were ground into particles smaller than 200 mesh and apportioned according to the calculations, and reactant samples were prepared by sufficient mixing. The thermodynamic calculation analyses were conducted for the ratios of raw materials in the samples.

**Table 2.** Composition of the charge for reduction roasting.

| Component | Hematite | CaO | C |
|---|---|---|---|
| Contents in wt.% | 75.09 | 8.79 | 16.11 |

## 3. Thermodynamic Calculation Analysis

During the reduction roasting of the samples, the possible chemical reactions included the reduction of iron oxide; generation of calcium ferrite by the combination of iron oxide and CaO; generation of tricalcium silicate (alite), $C_3S$ ($3CaO \cdot SiO_2$), and dicalcium silicate (belite), $C_2S$ ($2CaO \cdot SiO_2$), by the combination of CaO and $SiO_2$; and generation of $C_3S$ by the combination of CaO and $C_2S$. The equations for these reactions are shown in Equations (1)–(5). The reaction module of the FactSage software package (CRCT, Montreal, Canada; GTT-Technologies, Germany) was used to calculate the standard Gibbs free energies for all the possible reactions of the main phases in the reduction roasting process between 500 and 2000 °C at 10 °C intervals. Figure 2 shows the temperature (T) and standard Gibbs free energies ($\Delta G$) of the reactions.

$$Fe_2O_3 + 3\,C = 2\,Fe + 3\,CO \tag{1}$$

$$Fe_2O_3 + 2\,CaO = Ca_2Fe_2O_5 \tag{2}$$

$$2\,CaO + SiO_2 = Ca_2SiO_4 \tag{3}$$

$$3\,CaO + SiO_2 = Ca_3SiO_5 \tag{4}$$

$$CaO + Ca_2SiO_4 = Ca_3SiO_5 \tag{5}$$

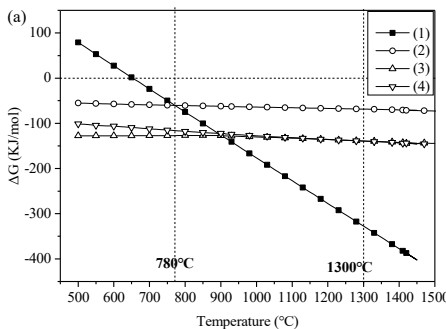
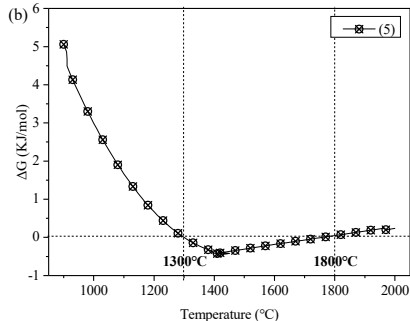

**Figure 2.** ΔG-T plots for the hematite coupling reactions. (**a**) Equations (1)–(4), (**b**) Equation (5).

Figure 2a shows that when the temperature was between 500 and 1450 °C, Equations (2)–(4) will proceed spontaneously, and tricalcium silicate, dicalcium silicate, and $Ca_2Fe_2O_5$ will be generated. The starting temperature for the reduction of iron in Equation (1) was 653 °C. However, when the temperature was below 780 °C, CaO and $Fe_2O_3$ were consumed, producing calcium ferrite through Equation (2); as this was more favorable than the reduction of iron through Equation (1), it was not conducive to the reduction of iron. When the temperature was above 780 °C, Equation (1) occurred before Equation (2), reducing $Fe_2O_3$ to iron. Therefore, a temperature above 780 °C is required for the reduction of $Fe_2O_3$.

By comparing the temperature versus standard Gibbs free energy curves for Equations (3) and (4), shown in Figure 2a, it can be seen that Equation (3) was more favorable than Equation (4) at temperatures below 1300 °C, and dicalcium silicate is easily generated. The formation of tricalcium silicate is more favorable when the temperature is above 1300 °C. It can also be seen from Figure 2b that Equation (5) only proceeds spontaneously when the temperature is between 1300 and 1800 °C, and tricalcium silicate can be generated through the combination of the dicalcium silicate and calcium oxide formed in the previous reactions. Figure 2a,b show that to generate cementing materials comprising predominantly tricalcium silicate, the temperature should be between 1300 and 1800 °C.

In order to investigate the phase composition of the reactants at high temperatures, when the iron was insufficiently reduced, the phase module of the FactSage software package was used to calculate and analyze the equilibrium phase diagram of the $Fe_2O_3$-CaO-$SiO_2$ ternary system at 1350 °C. This is shown in Figure 3.

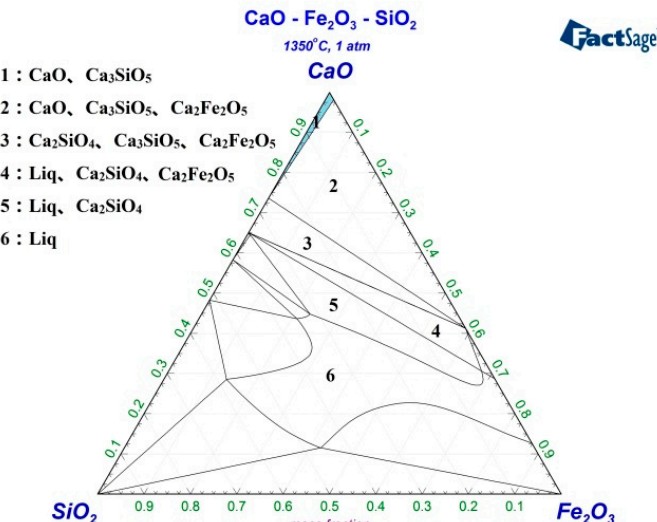

**Figure 3.** Equilibrium phase diagram of the $Fe_2O_3$-CaO-$SiO_2$ ternary system at 1350 °C.

Figure 3 shows that at 1350 °C, $Fe_2O_3$ reacts with CaO, generating $Ca_2Fe_2O_5$, and coexists with $C_2S$ and $C_3S$ in the high-CaO content regions (regions 2 and 3 in Figure 3); here, $Ca_2Fe_2O_5$ becomes an impurity in the cementing materials. When $Fe_2O_3$ has low concentrations or is absent (region 1 in Figure 3), the main phases present in the system are $C_3S$ and excess free CaO. Therefore, $Fe_2O_3$ should be sufficiently reduced to iron, by the roasting process, to prevent the formation of $Ca_2Fe_2O_5$. In the regions of high $Fe_2O_3$ and $SiO_2$ content (regions 4, 5, and 6 in Figure 3), there are low melting point liquid phases containing three components—namely, $Fe_2O_3$, CaO, and $SiO_2$—and $C_3S$ is not generated.

In order to study the interaction between the reduced iron and the CaO and $SiO_2$ in the system, the FactSage software package was used to calculate and analyze the phase diagram of the Fe-CaO-$SiO_2$ ternary system, in which the iron accounts for 50% of the total mass. These results are shown in Figure 4.

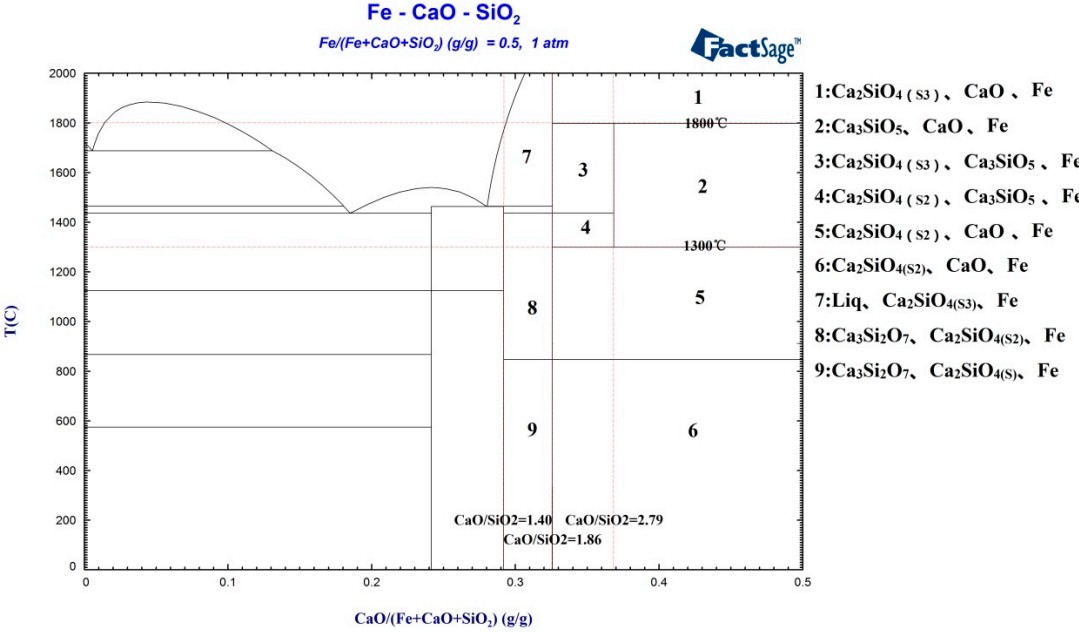

**Figure 4.** Phase diagram of the Fe-CaO-$SiO_2$ ternary system.

Figure 4 shows that tricalcium silicate is an unstable compound. Dicalcium silicate can form stable tricalcium silicate through combination with calcium oxide (regions 2, 3, and 4 in Figure 4) only within a temperature range of 1300–1800 °C; when the temperature is below 1300 °C or above 1800 °C (regions 1 and 5 in Figure 4), $C_3S$ decomposes into $C_2S$ and free CaO—this is in line with the thermodynamic calculation results given in Figure 2b. At the same time, it can also be seen that the reduced iron had no obvious effect on the CaO-$SiO_2$ system, meaning that iron can coexist with the CaO-$SiO_2$ system. When the concentration of CaO was such that CaO/$SiO_2$ < 1.40 and the temperature was between 1300 and 1800 °C, there were no $C_2S$ or $C_3S$ phases present in the system, and CaO mainly formed $CaSiO_3$ and $Ca_3Si_2O_7$ by combining with $SiO_2$. When CaO/$SiO_2$ was between 1.40 and 1.86, $C_2S$ was present in the system (regions 7 and 8 in Figure 4). When CaO/$SiO_2$ was between 1.86 and 2.79, $C_2S$ coexisted with $C_3S$ (regions 3 and 4 in Figure 4). When CaO/$SiO_2$ exceeded 2.79, the main phases in the system were $C_3S$, free CaO, and coexistent iron (region 2 in Figure 4). Thus, CaO/$SiO_2$ = 1.40 is the composition point for $Ca_3Si_2O_7$, CaO/$SiO_2$ = 1.86 is the composition point for $C_2S$, and CaO/$SiO_2$ = 2.79 is the composition point for $C_3S$. Therefore, when cementing materials with a tricalcium silicate main phase are being prepared, the masses of CaO and $SiO_2$ used in the blending process should have a ratio of CaO/$SiO_2$ = 2.79, and the roasting temperature should be between 1300 and 1800 °C.

The equipment module of the FactSage software package was used to perform a theoretical analysis of changes in the reactant phase composition during the temperature ramping, with hematite as the raw material, powdered carbon as the reducing agent, and CaO as the additive, under a protective

CO atmosphere. The ratio of the reactants used was as shown in Table 2. The phase change of the system between 500 and 1500 °C was analyzed, and the temperature difference was 50 °C. The Origin software package was used to prepare the plot. The theoretical reaction products, as determined by the calculations, are shown in Figure 5.

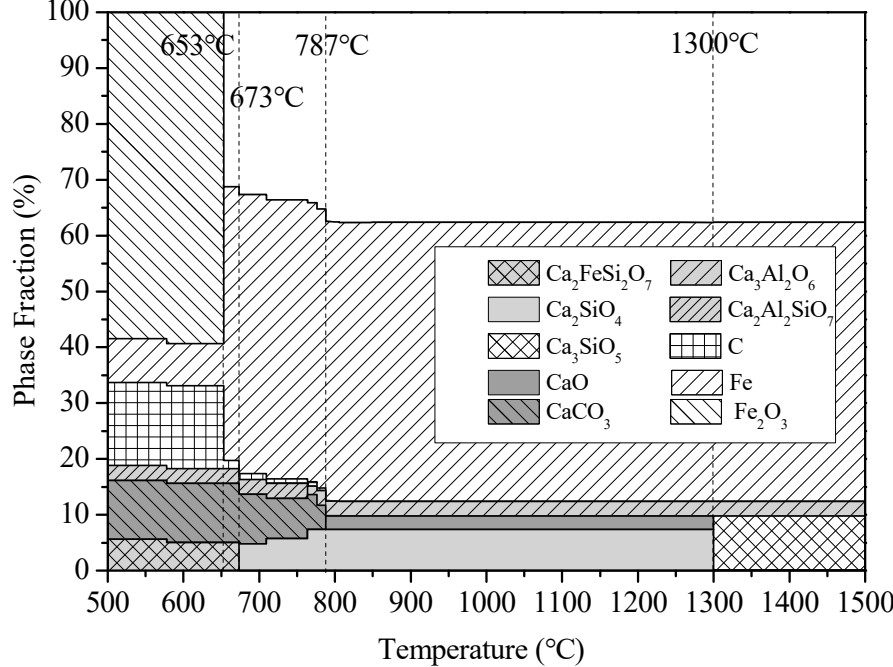

**Figure 5.** Variation diagram of the products of hematite reduction roasting.

From the calculation and analysis results shown in Figure 5, it can be seen that the reactions occur in the low-temperature region (<653 °C). The generated phases included $Ca_2Al_2SiO_7$, $CaCO_3$, $Ca_2FeSi_2O_7$, and a small amount of reduced iron, and there are large amounts of the reactants C and $Fe_2O_3$. When the temperature exceeds 653 °C, $Fe_2O_3$ is largely reduced by C to iron, and most of the reactants C and $Fe_2O_3$ are consumed. When the temperature is 673 °C, $C_2S$ is produced and the intermediate product, $Ca_2FeSi_2O_7$, is consumed. When the temperature is 787 °C, $CaCO_3$ completely decomposes into CaO and $CO_2$. When the temperature is between 787 and 1300 °C, the main phases are Fe, CaO, $C_2S$, and $Ca_3Al_2O_6$. When the temperature exceeds 1300 °C, CaO combines with $C_2S$ to form $C_3S$; the other phases do not change, and the final products, under high-temperature reaction conditions, are iron, $C_3S$, and $Ca_3Al_2O_6$.

Based on the above analysis, the preparation of elemental iron, $C_3S$, $C_2S$, and other cementing materials by reduction roasting of hematite, after apportioning the reactants according to the thermomechanical analysis, is feasible. At the same time, in order to ensure sufficient reduction of the iron and effective formation of the cementing materials (such as $C_3S$ and $C_2S$), the reaction temperature needs to be controlled in stages.

In order to prevent $Fe_2O_3$ and CaO combining to form $Ca_2Fe_2O_5$, which affects the reduction of iron, the reaction temperature, for $Fe_2O_3$, should be between 780 and 1220 °C. This temperature should be maintained for a sufficient time to fully reduce the iron and prevent formation of the low melting point liquid phase (the melting point of fayalite is 1220 °C) from $Fe_2O_3$ and $SiO_2$. By this stage, $C_2S$ has been generated by the combination of CaO and $SiO_2$. To achieve sufficient reduction of the iron, the temperature should be raised for a second time, and $C_3S$ generated through the reaction of $C_2S$ with CaO (1300–1800 °C). Finally, the samples should be cooled by fast cooling. During fast cooling, the decomposition rate of tricalcium silicate is slow enough that it can exist as a metastable state at ambient temperature.

## 4. Analysis of Experimental Results

In this experiment, to verify the effects of the different reaction conditions on the results of the thermodynamic calculations, three different control processes were designed, as follows: (1) 8 g of sample was placed in a corundum crucible in an airtight, box-type resistance furnace. Without a protective atmosphere, the temperature was continuously raised from room temperature to 1450 °C at a ramp rate of 10 °C/min; the product, A, was obtained after heating for 1 h and cooling to room temperature. (2) 8 g of sample was placed in a corundum crucible and then, placed in a tube-type resistance furnace. Under a protective CO atmosphere, with a flow rate of 60 mL/h, the temperature was continuously raised from room temperature to 1000 °C and, after heating for 2 h, continuously raised to 1450 °C at a ramp rate of 10 °C/min; the product, B, was obtained after heating for 1 h at 1450 °C and cooling to room temperature. (3) 8 g of sample was reduced and roasted, and the same atmosphere and temperature ramp to 1450 °C were used as for sample B; the difference was that, after heating for 1 h at 1450 °C, the product, C, was obtained using the fast cooling method. Finally, XRD and scanning electron microscopy–energy-dispersive X-ray spectroscopy (SEM–EDS) analyses were conducted on the samples obtained. The same diffractometer was used as for the raw material analysis; the scanning electron microscope used was a VEGA II XMU from TESCAN, Czech Republic, and the EDS was type 7718 from Oxford Instruments, UK. The analysis results are shown in Figures 6 and 7.

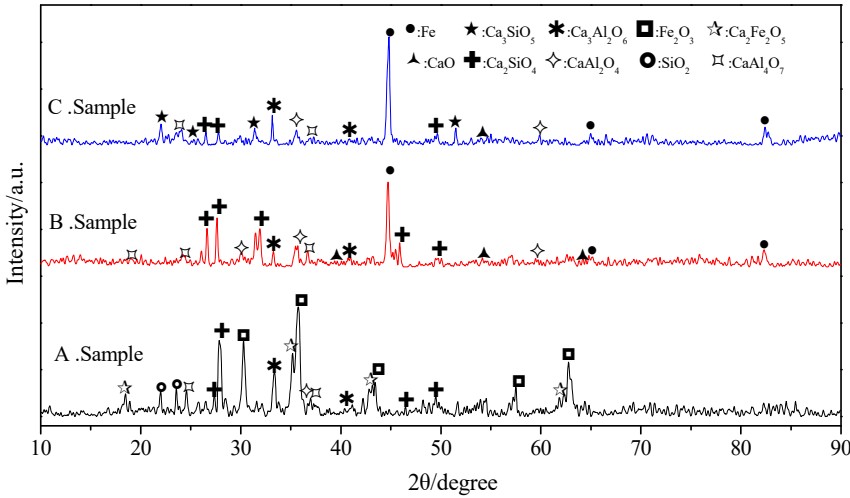

**Figure 6.** XRD patterns of hematite reduction roasting products.

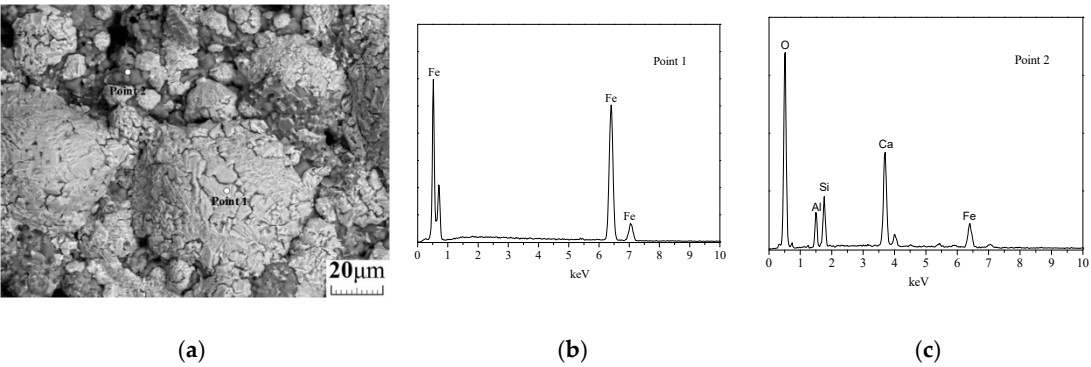

(**a**)　　　　　　　　(**b**)　　　　　　　　(**c**)

**Figure 7.** SEM–EDS results for sample C. (**a**) surface topography, (**b**) elementary composition of point 1, (**c**) elementary composition of point 2.

The iron oxide accounted for 70.58% of the mass of the sample, and the elemental iron in the theoretical reaction product accounted for 80.27% of the total mass. Therefore, strong peaks from elemental iron and iron oxide appear in the XRD patterns of the samples after the reaction, and the X-ray diffraction peaks of the other phases are generally low.

From the results of the XRD analysis, it can be seen that, for sample A, the reduction of iron could not be ensured, due to the lack of a protective atmosphere; the iron was present in the form of $Fe_2O_3$, which formed $Ca_2Fe_2O_5$ through combination with CaO. The consumption of CaO resulted in reductions in the $CaO/SiO_2$ and $CaO/Al_2O_3$ ratios. The X-ray diffraction peaks of phases with low calcium contents, such as $CaAl_4O_7$ and $Ca_2SiO_4$, were obvious in the reaction products, and diffraction peaks of $SiO_2$ and $Ca_3Al_2O_6$ were also present, which was consistent with the analysis results shown in Figure 3. With the protection of a reducing atmosphere, sample B was heated at 1000 °C for 2 h, ensuring the complete reduction of iron, and then, at 1450 °C for 1 h, matching the formation conditions of $C_3S$; however, $C_3S$ decomposed into $C_2S$ and CaO phases, due to the use of furnace cooling. Thus, the final reaction products of sample B were phases such as elemental iron, $Ca_2SiO_4$, CaO, $CaAl_4O_7$, $CaAl_2O_4$, and $Ca_3Al_2O_6$; this was consistent with the analysis results shown in Figure 4. Under a protective CO atmosphere, sample C was heated according to the same protocol as sample B, to realize the reduction of iron and formation of $C_3S$ and $C_3A$ phases, and fast cooling was used to avoid large-scale decomposition of $C_3S$. The final reaction products were Fe, $C_3S$, and $Ca_3Al_2O_6$, and the decomposition products were $Ca_2SiO_4$ and $CaAl_2O_4$, which showed substantially weaker X-ray diffraction peaks than the $Ca_2SiO_4$ in sample B. At the same time, from the results of the SEM–EDS analysis of sample C, it can be seen that the product morphology consisted of many iron grains, with different diameters and cementing materials among them; this was consistent with the analysis results of the theoretical products shown in Figure 5.

## 5. Conclusions

(1)  It is feasible to prepare elemental iron, $C_3S$, $C_2S$, and other cementing materials that can coexist at high temperatures by reduction roasting of hematite after apportioning the reactants using thermomechanical analysis.

(2)  The coupling reaction temperature needs to be controlled in stages. The reduction temperature of $Fe_2O_3$ should be between 780 and 1220 °C, and the formation temperature of $C_3S$ should be between 1300 and 1800 °C. In order to obtain a stable $C_3S$ phase, the fast cooling method must be used for sampling.

(3)  Under the protective, reducing CO atmosphere, the temperature of the first stage is 1000 °C, and the heating time is 2 h; the temperature of the second stage is 1450 °C, and the heating time is 1 h. The fast cooling method is used for sampling; the final products of the hematite reduction roasting are Fe, $C_3S$, and $Ca_3Al_2O_6$, and the decomposition products are $Ca_2SiO_4$ and $CaAl_2O_4$.

(4)  The reduction of iron has a great influence on the formation of $C_3S$. When iron has not been sufficiently reduced, it will form calcium ferrite, and other phases, through combination of the CaO and $SiO_2$; this hinders the formation of $C_3S$. Sufficient reduction of the iron is a precondition for the formation of $C_3S$.

**Author Contributions:** Methodology, R.S., J.Z., X.L., C.Z. and Y.C.; software, J.Z., X.L., C.Z. and Y.C.; validation, J.Z. and X.L.; R.S., formal analysis, C.Z. and Y.C.; investigation, R.S., J.Z., X.L., C.Z. and Y.C.; data curation, C.Z. and Y.C.; writing—original draft preparation, R.S., J.Z.; writing—review and editing, J.Z. and G.Q. All authors have read and agreed to the published version of the manuscript.

**Funding:** The authors wish to express their thanks to the National Natural Science Foundation of China (51874224) and Project of young talents in basic research of Natural Science in Shaanxi Province (2014JQ7282) for the financial support of this research.

**Conflicts of Interest:** The authors declare no conflict of interest.

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
