# Peer review of "Experimental Study on the Preparation of Cementing Materials by Direct Reduction Coupling of a Hematite-Carbon Base"

_metals, doi:10.3390/met10081086_

Round 1

Reviewer 1 Report

Dear Authors, your work is interesting.

1. The analysis of the available knowledge is very good.

2. The idea to do research is good.

3. The research methodology is correct.

4. The obtained results are correctly described.

5. Numerical analyzes (FactSage software) complete the research.

6. On lines 102-106, the word Equation is unnecessarily repeated. It is enough to mark (1), (2) etc.

In my opinion, the article is very well written and is suitable for publication in the journal Metals.

Yours faithfully
reviewer

Reviewer 2 Report

The manuscript is interesting, however, requires significant corrections before publications.

Line 67. Add the number of crystallographic card used, corresponding to the hematite.

Line 72. Delete the text. “The main component of the hematite powder was Fe2O3.” Because this idea was discussed on line 69-70.

Line 73. In table captions from table 1, “Chemical composition of the hematite powder (1%)”, ¿ why is it 1%? In the same table in components "TFe" appears, perhaps it should be Fe2O3. ¿Why do the sum of the components is 71.051% and not equal or approximate 100%?

Line 74. I suggest changing the statement, "From the analysis of the fluorescence spectrum", to "From Table 1 (values of the analysis of the fluorescence spectrum),".

Line 91. Why does the apportioning of hematite, CaO and C, give those values?

Line 99-100. The statement, “Additionally, Origin was used to plot the relationship between the temperatures and standard Gibbs free energies of the reactions; these results are shown in Figure 2.”, I suggest changing the statement by, “Figure 2, shows the temperature (T) and standard Gibbs free energies (ΔG) of the reactions.”

Line 107. It is necessary to identify each figure with the letters (a) and (b). Both figures on the x-axis are the word "Tempreature (°C)", change it to "Temperature (°C)". In the figure on the left (Fig. 2 (a)), the dotted line should appear at 1300 ° C.

Line 144. In Figure 4, in the y-axis, it's not degrees (°).  En el mismo gráfico, la relación CaO / SiO2, debe ser, CaO / SiO2.

Line 68. Delete the statement. “The Origin software package was used to prepare the plot”.

Line 216. On the micrograph highlight the words "Point 1" and "Point 2".

Reviewer 3 Report

Fig 1. shows hematite XRD pattern and Table 1 also shows chemical composition of that. But It does not show Fe2O3 composition but FeO. Revise it.

There is no result and discussion on the reduction rate and/or recovery rate of the specimen. Add the comments in the article.

Thank you.

Reviewer 4 Report

1.

Table 2. Apportioning of hematite reduction roasting.

 Rather     - composition of the charge for reduction roasting

 90 Table 2. Apportioning of hematite reduction roasting.

Component      Hematite     CaO         C

Proportion         75.09            8.79     16.11

 Proportion – Contents in % wt.

2

 Alit – C3S (Ca3SiO5)

 of tricalcium silicate, C3S (alite), and dicalcium silicate, C2S (belite), by the combination of 96 CaO and SiO2; and generation of C3S by the combination of CaO and C2S.

  dicalcium silicate 2CaO·SiO2 - C2S (belite)

Metallurgical chemical formulas instead of those used in the cement industry should be appropriate

3.

Figure 2. ΔG-T plots for the hematite coupling reactions.

change the Gibbs Enthalpy units     J/mol to kJ/mol

  1.  

There is no discussion on:

  • Which phase Alit or Belit is more important in cement production? Belite is the mineral in Portland cement responsible for development of "late" strength. The other silicate, alite contributes "early" strength, due to its higher reactivity. Belite reacts with water (roughly) to form calcium silicate hydrates (C-S-H) and portlandite (Ca(OH)2).
  • What is structure and composition of iron and “slag”- cement components ?
  • What about  self disintegration of 2CaO·SiO2 very  important for cement production?
  • What you mind  CO protective atmosphere? protective against what?

Reviewer 5 Report

The paper is interesting and well organized. The problem of unconventional methods of cement-like binders production is important.

The description of the experiment is very precise and justifies the conclusions.

Few questions for the authors:

- composition of the tested material in table 1 is not complete - summ of the components is only ca. 70% by mnass, what are the other compounds?

- waht means "(1%)" in the title of table 1?

- any try to test the obtained material as a binder? I would like to know how this material reacts with water, waht are the technical and chemical effects of this reaction - is i t possible to use the new "cement" for any practical purposes?

Abovementioned remarks are only for help authors to improve the paper

In my opinion the paper is almost ready for publication
